# Digital access, transportation, and women's empowerment in breast cancer screening uptake among Cambodian women: Analysis of the Cambodia demographic and health survey 2021–2022

**Samnang Um**[1]*, **Channnarong Phan**[1], **Daraden Vang**[1], **Tharuom Ny**[2], **Sothy Heng**[3]

**1** National Institute of Public Health, Phnom Penh, Cambodia, **2** Khmer Soviet Friendship Hospital, Phnom Penh, Cambodia, **3** Ministry of Health, Phnom Penh, Cambodia

* umsamnang56@gmail.com

## Abstract

Breast cancer incidence is increasing globally, and it is the third leading cause of morbidity and mortality among women in Cambodia. This study explores how access to digital tools, media exposure, transportation, travel time to health facilities, and autonomy in health decisions relate to breast cancer screening among Cambodian women aged 15–49. The study used nationally representative, cross-sectional data from the Cambodia Demographic and Health Survey (CDHS) 2021–2022. After excluding 204 women who were unaware of breast or cervical cancer screening, the final weighted sample comprised 19,292 participants. The outcome was whether a woman had ever received a breast examination from a healthcare provider, encompassing clinical breast examinations (CBEs) and imaging techniques, such as mammograms. Multivariable logistic regression, adjusted for demographic and socio-economic characteristics, was used. Only 10.9% (95% CI: 9.7%–11.6%) of women had undergone a breast exam. Exposure to multiple forms of media was associated with a higher odds of screening (AOR = 1.47; 95% CI: 1.13–1.91). Phone ownership—both non-smartphone (AOR = 1.35; 95% CI: 1.03–1.78) or smartphone (AOR = 1.37; 95% CI: 1.03–1.82)—was also positively associated. In contrast, longer travel times of over 30 minutes (AOR = 0.55; 95% CI: 0.39–0.78) and a lack of autonomy in healthcare decisions (AOR = 0.70; 95% CI: 0.52–0.94) were associated with reduced screening. Wealthier women had greater odds of being screened (AOR = 1.86; 95% CI: 1.40–2.48). These findings highlight the need for health initiatives that use digital communication to reach and emphasize the importance of improving transportation, and support women's decision-making to increase screening rates in Cambodia.

**Data availability statement:** This study used the 2021-2022 Cambodia Demographic and Health Survey (CDHS) datasets. The DHS data are publicly available from the website at (URL: https://www.dhsprogram.com/data/available-datasets.cfm).

**Funding:** The author(s) received no specific funding for this work.

**Competing interests:** The authors have declared that no competing interests exist.

**Abbreviations:** ACS, American Cancer Society; AOR, Adjusted odds ratio; BMI, Body mass index; CDHS, Cambodia Demographic Health Survey; EA, Enumeration areas; HPV, Human papillomavirus; NCDs, Noncommunicable diseases; PPS, Probability proportional to size; VIA, Visual inspection with acetic acid; WRA, Women at reproductive age; WHO, World Health Organization.

## Author summary

Breast cancer is a significant health issue in Cambodia, but the percentage of screening remains low. This study explores how access to digital tools, media exposure, transportation, travel time to health facilities, and autonomy in health decisions could influence breast screening behavior. This study utilized the latest nationally representative and cross-sectional data from the Cambodia Demographic and Health Survey (CDHS) 2021–2022, which included 19,292 weighted women aged 15–49. This study found that only about 11% of women had ever had a breast exam. Women who owned a mobile phone (non-smart phones and smartphones) and were exposed to more media (including newspapers, radio, and television) were more likely to have had a breast exam. In contrast, we found that women who lived more than 30 minutes from a health facility or who were not autonomous in their healthcare decisions were less likely to be screened. These findings highlight the need for health initiatives that use digital communication to reach women and emphasize the importance of improving transportation and supporting women's decision-making to increase breast cancer screening rates in Cambodia.

## Introduction

In 2022, the Global Cancer Observatory estimated approximately 24% of new cases (2,296,840 new breast cancer cases), out of a total of 9,664,889 cancer cases reported in 2022, and 666,103 (or 15.4%) deaths due to breast cancer globally, represented a significant public health burden [1]. The rising burden is particularly in low- and middle-income countries (LMICs), including Cambodia, where late-stage diagnosis, limited access to screening, and underdeveloped cancer control infrastructure contribute to poorer outcomes [2–4].

In Cambodia, breast cancer accounts for a significantly increased rate of breast cancer morbidity and mortality, accounting for approximately 20% of new cases (2,116 new female cancer cases), out of a total of 10,624 cancer cases reported in 2022, and 917 (or 14%) deaths due to breast cancer [5]. Despite limited national data, hospital registries and regional studies show rising incidence [3,5–10]. Early detection through breast cancer screening—especially clinical breast examination (CBE)— is a cost-effective and feasible method in resource-limited settings and improves survival [4,11–19]. Yet, coverage remains low in Cambodia, with only about 11% of women aged 15–49 having ever had a breast exam performed by a health professional, with persistent socioeconomic, cultural, and geographic barriers impeding access [3–5,7]. In addition, recent studies indicated that knowledge about breast self-examination (BSE) in Cambodia was limited, with approximately 60% of Cambodian women being aware of the BSE method [3]. To address this, strengthening cancer screening in the Royal Government of Cambodia (RGC) requires ongoing efforts.

Cambodia MoH The National Action Plan for Cervical Cancer Prevention and Control, 2019–2023, has prioritized it, which emphasizes strengthening awareness campaigns, expanding access to screening, and referral systems. Progress included pilot programs integrating cervical and breast cancer awareness into community campaigns, though nationwide coverage remained limited. More recently, the National Cancer Control Plan (2025–2030) has prioritized decentralizing services, adopting digital health approaches, and piloting the Phnom Penh Population-Based Cancer Registry [13,17].

Recent technological developments offer new opportunities to enhance awareness and uptake of screening services. In 2022, mobile phone ownership is increasing among Cambodians, exceeding 80%, and internet penetration is expanding rapidly [6]. Social Media exposure, such as television, radio, and newspapers, remains a powerful channel for public health messaging and outreach interventions within the community, especially in LMICs [7,8]. Studies from other settings have shown that women with digital access and frequent media exposure can improve their cancer knowledge and promote screening behaviors [9,10]. However, these relationships have not been comprehensively studied in the Cambodian context.

Transportation is another key determinant. Motorcycles—the dominant mode of transport in low-middle income Cambodia, particularly in rural and remote areas—can reduce travel barriers and increase women's independence [11]. Motorcycle ownership may enable women to overcome geographic barriers by reducing travel time and increasing independence in reaching health services [12]. Yet, its specific role in facilitating breast cancer screening has not been systematically examined.

Barriers to healthcare access—such as distance to facilities, financial constraints, permission from others, or unwillingness to seek care alone—also remain prevalent [13]. Overall, women aged 15–49 years reported a higher proportion, 60.4%, having at least one barrier to accessing healthcare for themselves in Cambodia [20]. Women's autonomy in healthcare decision-making is another critical factor that may influence the uptake of preventive services like cancer screening, but it remains understudied in Cambodia [20].

A previous study using CDHS 2021–2022 data has examined the socio-demographic and behavioral determinants of breast and cervical cancer screening in Cambodian women aged 15–49 years [14]. While that analysis did not explore the influence of emerging digital access, transportation, travel time to the facility, and women's decision-making autonomy [14]. This study addresses these knowledge gaps by utilizing the latest nationally representative and cross-sectional data from the CDHS 2021–2022 to examining how digital access (ownership mobile phone type, media exposure, and internet use), motorcycle ownership, travel time to the facility, and women's decision-making autonomy are associated with breast cancer screening, adjusting for socio-demographics factors.

## Methods

### Ethical statement

The Cambodia National Ethics Committee for Human Health Research (NECHR) approved the data collection tools and procedures for CDHS 2021–2022 for Health Research on May 10, 2021 (Ref **# 83 NECHR**), and ICF's Institutional Review Board (IRB) in Rockville, Maryland, USA. Written informed consent was obtained from all participants before data collection. For respondents under 18 years of age, consent was obtained from a parent or guardian. This study used de-identified secondary data and was therefore exempt from additional institutional ethical approval.

### Data source

This study utilized data from the Cambodia Demographic and Health Survey (CDHS) 2021–2022, a nationally representative cross-sectional survey designed to provide estimates of key demographics, reproductive health, and nutrition indicators for women and children. Data was collected from September 15, 2021, to February 15, 2022. The CDHS employed a two-stage stratified cluster sampling method, selecting women from all provinces of Cambodia. In the initial stage, 709 enumeration areas (EAs) (241 urban areas and 468 rural areas) were selected. In the second stage, an equal systematic

sample of 25–30 households was selected from each cluster of 21,270 families. In total, 19,496 women aged 15–49 were interviewed face-to-face, using the survey questionnaire, with a response rate of 98.2%. The final CDHS 2021–2022 report details have been published [8].

## Study population

The analysis included women aged 15–49 who responded to questions on breast cancer screening performed by a health professional, yielding 19,292 weighted observations. Women with missing data on the outcome variables (n = 204) were excluded.

## Measurement variables

**Outcome variable.** The outcome was breast cancer screening, measured by asking women if they had ever had their breasts examined by a doctor or other health care provider to check for cancer. This examination encompassed both clinical breast examinations (CBE), where providers manually palpate for lumps or changes, and imaging techniques, such as mammograms [8]. This variable was coded as binary (1 = Yes, 0 = No) [8].

**Independent variables. Digital Access:** This was measured by three separate variables: type of mobile phone ownership (no phone, non-smartphone, and smartphone), internet use within the past 12 months (yes/no), and media exposure measured by the frequency of reading newspapers, listening to the radio, and watching television, then were categorized into (none, exposure to one source, or exposure to two or more sources) (8).

**Transportation:** This was measured by household motorcycle ownership (yes vs. no).

**Healthcare access:** This included the time required to travel to the nearest health facility, categorized as near (≤10 minutes), moderate (11–30 minutes), or far (>30 minutes).

**Women's autonomy in healthcare decision-making**: This was categorized into three groups based on who makes decisions regarding the respondent's healthcare: autonomous (respondent alone), joint (shared decision-making), and non-autonomous (decisions made by others).

**Covariate variables** included women's age (15–29, 30–39, and 40–49 years), education level (no education, incomplete primary, complete primary, incomplete secondary, complete secondary, and higher), and household wealth status (classified into quintiles from poorest to richest) [8]. Additional geographic variables included place of residence (urban or rural) [8].

## Statistical analysis

All analyses accounted for the complex survey design, utilizing sampling weights, clustering, and stratification as provided by the CDHS. Descriptive statistics summarized sample characteristics and breast cancer screening coverage, reported as weighted counts and percentages.

Bivariate associations were tested using chi-square tests, with weighted frequency proportions and p-values based on survey-adjusted F-statistics, and unadjusted logistic regression models. Multivariable logistic regression with survey design adjustment was used to examine factors associated with breast cancer screening. The analysis accounted for the complex survey design and reported design-based F-statistics, including stratification, clustering, and sampling weights using the **svy**: prefix in STATA (StataCorp, 2023) [21]. The following independent variables were included: media exposure, smartphone ownership, internet use in the last 12 months, motorcycle ownership, time to reach health care, decision-making autonomy, current age group, household wealth index, education level, occupation, and place of residence. The final results of the multivariable binary logistic regression were reported as adjusted odds ratios (AOR) with 95% confidence intervals (CI), and p-values. Statistical significance was defined as $p < 0.05$.

The goodness-of-fit was assessed using the Hosmer-Lemeshow test with ten groups [22]. Predicted probabilities were generated post-estimation to evaluate model discrimination via the area under the receiver operating characteristic (ROC) curve [22]. Statistical significance was determined at a p-value <0.05.

Multicollinearity among the independent variables was assessed using Variance Inflation Factors (VIFs) derived from a linear regression model that included all predictors [23]. A VIF value greater than 5 was considered indicative of problematic multicollinearity [23]. Variables included in the multicollinearity assessment were media exposure, smartphone ownership, internet use, motorcycle ownership, decision-making group, women's age, wealth index, education, occupation, and place of residence. The VIF values for all predictors ranged from 1.01 to 1.79, with a mean VIF of 1.33, indicating low multicollinearity among the independent variables. No variable exceeded the threshold of 5, suggesting that multicollinearity was not a concern in the model [23].

We assessed potential effect modification between smartphone ownership and age by including an interaction term in the multivariable logistic regression model using the factor-variable notation, as these reflect plausible conceptual links between access and empowerment. Other potential interactions, such as media exposure and education, and smartphone and residence, wealth, and media exposure, were examined. The significance of interaction terms was evaluated using adjusted Wald tests with the testparm command under the complex survey design [21].

## Results

### Description of the study samples

Among 19,292 weighted women aged 15–49 years, the average age was 31.0 years (95% CI: 30.9–31.2). The majority had no mass media exposure (71.9%), owned a smartphone (77.5%), and had used the internet in the past year (63.6%). Over half (60.8%) owned a motorcycle, and half (50.8%) lived within a 10-minute distance of a health facility. About 44% reported autonomy in healthcare decision-making, with 23.1% in the richest and 17.4% in the poorest group. Most women had incomplete secondary education (35.6%) or incomplete primary education (29.3%). Approximately 42.4% worked in informal jobs, while 30.4% held professional roles. A larger proportion lived in rural areas (57.5%). Overall, only 10.9% (95% CI: 9.7%-11.6%) of women reported ever having a breast exam by health providers (Table 1).

### Association with breast cancer screening in Chi-square

The prevalence of breast cancer screening (ever having a breast exam by a health professional) was 10.6% (95% CI: 9.7%–11.6%). Women who had undergone a breast exam were, on average, older, with a mean age of 35.2 years (95% CI: 34.6–35.7), compared to 30.6 years (95% CI: 30.4–30.7) among those who had not. Screening prevalence increased with age: 5.5% among women aged 15–29 years, 14.7% among those aged 30–39 years, and 14.4% among women aged 40–49 years. These differences in breast exam uptake across age groups were statistically significant ($F_{(1.97, 1301.28)} = 74.6$; $p < 0.001$). Women's exposure to mass media was significantly associated with breast exam uptake ($F_{(1.82, 1203.01)} = 16.4$; $p < 0.001$). The proportion of women who had ever undergone a breast exam increased from 9.5% among those with no media exposure to 16.2% among those exposed to two or more sources. Smartphone ownership and recent internet use were also associated with higher uptake ($F_{(2.00, 1319.60)} = 11.9$; $p < 0.001$ and $F_{(1, 660)} = 5.81$; $p = 0.016$, respectively). Specifically, 11.2% of smartphone users had ever undergone a breast exam, compared to 7.2% of women who did not own a smartphone. Access to healthcare services was a crucial factor in this decision. Uptake was highest among women living within 10 minutes of a health facility (12.3%) and lowest among those living more than 30 minutes away (5.0%) ($F_{(1.80, 1187.66)} = 21.8$; $p < 0.001$). Similarly, women with autonomy in healthcare decision-making had a higher uptake (15.1%) compared to those without (10.7%) ($F_{(1.96, 1295.40)} = 63.9$; $p = 0.0015$). Breast exam uptake increased with household wealth, ranging from 7.3% in the poorest quintile to 17.7% in the richest ($F_{(2.96, 1954.75)} = 31.3$; $p < 0.001$). A similar pattern was seen by education level; women with higher education had

PLOS Digital Health

**Table 1. Distribution of Characteristics of Media Access, Digital Access, Motorcycle Access, Healthcare decision-making, and Socioeconomic Status Associated with Breast Exam Uptake Among Women Aged 15–49 in Cambodia, CDHS 2021–2022 (N = 19,292 weighted).**

| Variable | Total | Breast Exam Uptake | | F-statistics | P-value |
|---|---|---|---|---|---|
| | | Yes | No | | |
| | N (%) | n (%) | n (%) | | |
| **Media Access** | | | | | |
| None | 13,865 (71.9) | 1,321 (9.5) | 12,544 (90.5) | 16.37 | <0.001 |
| One source | 4,433 (23.0) | 564 (12.7) | 3,869 (87.3) | | |
| Two or more sources | 993 (5.1) | 161 (16.2) | 832 (83.8) | | |
| **Smartphone Ownership** | | | | | |
| No phone | 2,917 (15.1) | 210 (7.2) | 2,707 (92.8) | 11.90 | <0.001 |
| Non-smartphone | 1,423 (7.4) | 156 (11.0) | 1,266 (89.0) | | |
| Smartphone | 14,952 (77.5) | 1,679 (11.2) | 13,273 (88.8) | | |
| **Internet use (last 12 months)** | | | | | |
| No | 7,020 (36.4) | 661 (9.4) | 6,359 (90.6) | 5.81 | 0.016 |
| Yes | 12,272 (63.6) | 1,385 (11.3) | 10,887 (88.7) | | |
| **Motorcycle Ownership** | | | | | |
| No | 7,570 (39.2) | 932 (12.3) | 6,638 (87.7) | 0.30 | 0.581 |
| Yes | 11,721 (60.8) | 1,113 (9.5) | 10,608 (90.5) | | |
| **Travel time to the facility** | | | | | |
| ≤10 minutes | 9,811 (50.8) | 1,208 (12.3) | 8,602 (87.7) | 21.75 | <0.001 |
| 11–30 minutes | 8,397 (43.5) | 783 (9.3) | 7,614 (90.7) | | |
| >30 minutes | 1,084 (5.6) | 54 (5.0) | 1,030 (95.0) | | |
| **Healthcare decision-making** | | | | | |
| Autonomous | 5,866 (30.4) | 4,983 (84.9) | 883 (15.1) | 63.92 | <0.001 |
| Joint | 6,379 (33.1) | 5,619 (88.1) | 760 (11.9) | | |
| Not autonomous | 7,043 (36.5) | 6,641 (94.3) | 402 (5.7) | | |
| Women's mean age in years (95% CI) | 31.0 (30.9–31.2) | 35.2 (34.6–35.7) | 30.6 (30.4–30.7) | | |
| 15-29 | 8,444 (43.8) | 464 (5.5) | 7,980 (94.5) | 74.60 | <0.001 |
| 30-39 | 6,589 (34.2) | 967 (14.7) | 5,622 (85.3) | | |
| 40-49 | 4,258 (22.1) | 614 (14.4) | 3,644 (85.6) | | |
| **Wealth index** | | | | | |
| Poorest | 3,355 (17.4) | 246 (7.3) | 3,108 (92.7) | 31.32 | <0.001 |
| Poorer | 3,451 (17.9) | 262 (7.6) | 3,189 (92.4) | | |
| Middle | 3,790 (19.6) | 314 (8.3) | 3,476 (91.7) | | |
| Richer | 4,243 (22.0) | 434 (10.2) | 3,808 (89.8) | | |
| Richest | 4,453 (23.1) | 789 (17.7) | 3,664 (82.3) | | |
| **Education level** | | | | | |
| No education | 2,232 (11.6) | 197 (8.8) | 2,035 (91.2) | 4.98 | 0.0005 |
| Incomplete primary | 5,652 (29.3) | 608 (10.8) | 5,044 (89.2) | | |
| Complete primary | 1,823 (9.4) | 204 (11.2) | 1,619 (88.8) | | |
| Incomplete secondary | 6,874 (35.6) | 654 (9.5) | 6,220 (90.5) | | |
| Complete secondary | 1,320 (6.8) | 168 (12.7) | 1,153 (87.3) | | |
| Higher | 1,390 (7.2) | 215 (15.5) | 1,175 (84.5) | | |
| **Occupation** | | | | | |
| Not working | 5,249 (27.2) | 427 (8.1) | 4,823 (91.9) | 37.02 | <0.001 |
| Professional | 5,866 (30.4) | 896 (15.3) | 4,970 (84.7) | | |
| Informal | 8,176 (42.4) | 723 (8.8) | 7,453 (91.2) | | |

*(Continued)*

**Table 1.** (Continued)

| Variable | Total | Breast Exam Uptake | | F-statistics | P-value |
| --- | --- | --- | --- | --- | --- |
| | | Yes | No | | |
| | N (%) | n (%) | n (%) | | |
| **Place of residence** | | | | | |
| Urban | 8,205 (42.5) | 1,075 (13.1) | 7,131 (86.9) | 22.14 | <0.001 |
| Rural | 11,086 (57.5) | 971 (8.8) | 10,115 (91.2) | | |

Survey weights are applied to obtain weighted percentages.

greater uptake (15.5%) compared to those with no education (8.8%) (F(4.12, 2717.74) = 4.98; p = 0.0005). Women in professional occupations were more likely to have had a breast exam (15.3%) than those not working (8.1%) or working in informal jobs (8.8%) (F(1.95, 1284.05) = 37.0; p < 0.001). Notably, urban women reported higher uptake (13.1%) than rural women (8.8%) (F(1, 660) = 22.1; p < 0.001), contrary to typical patterns in healthcare access (Table 1).

### Factors associated with breast exam uptake in multivariable analysis

The final multivariable binary logistic regression model was statistically significant (F(24, 637) = 8.71, p < 0.001), indicating that the set of predictors explained a substantial amount of variation in breast cancer screening.

In the fully adjusted model, several factors remained significantly associated with breast exam uptake among women aged 15–49 years. As shown in Table 2, women exposed to mass media had higher odds of breast exam uptake compared to those with no media exposure; those with one source had 1.34 times the odds (AOR = 1.34; 95% CI: 1.13–1.58), and those with two or more sources had 1.47 times the odds (AOR = 1.47; 95% CI: 1.13–1.91). Similarly, phone ownership was positively associated with screening uptake. Both non-smartphone users (AOR = 1.35; 95% CI: 1.03–1.78) and smartphone users (AOR = 1.37; 95% CI: 1.03–1.82) had significantly higher odds of breast exam uptake compared to women without phones. Longer travel time to the nearest health facility was inversely related to breast exam uptake: women living 11–30 minutes away had 16% lower odds (AOR = 0.84, 95% CI: 0.71–0.98), and those living more than 30 minutes away had 45% lower odds (AOR = 0.55, 95% CI: 0.39–0.78), compared to women living within 10 minutes. Healthcare decision-making autonomy was also significant; women who made decisions jointly with others (AOR = 0.82, 95% CI: 0.70–0.96) or were not autonomous (AOR = 0.70, 95% CI: 0.52–0.94) had significantly lower odds of screening compared to women who made healthcare decisions autonomously. Age remained a strong predictor of screening uptake. Women aged 30–39 years had 1.8 times the odds (AOR = 1.8; 95% CI: 1.49–2.23), and those aged 40–49 years had twice the odds (AOR = 2.04, 95% CI: 1.61–2.58) of undergoing a breast exam compared to women aged 15–29 years. Regarding socioeconomic status, only women in the richest wealth quintile had significantly higher odds of screening (AOR = 1.86, 95% CI: 1.40–2.48) compared to those in the poorest quintile. Educational attainment was associated with uptake only at the level of completing secondary school (AOR = 1.47, 95% CI: 1.04–2.08). Occupation and urban versus rural residence were not statistically significant after adjustment for other factors. Internet use and motorcycle ownership did not show significant associations in the adjusted model (Table 2).

The Hosmer-Lemeshow goodness-of-fit test indicated adequate model fit (F(9,652) = 0.42, p = 0.925). The ROC curve analysis yielded an area under the curve (AUC) of 0.653 (95% CI: 0.638–0.668), indicating moderate discriminative ability.

The interaction between smartphone ownership and age was statistically significant (Adjusted Wald test: F(8, 653) = 9.54, p < 0.001), indicating that the association between smartphone ownership and breast cancer screening varied across age groups. For example, among women aged 40–49, smartphone users had significantly higher odds of screening (AOR = 2.22; 95% CI: 1.34–3.68; p = 0.002), compared to non-users aged 15–29. Similarly, among those aged

**Table 2. Unadjusted and adjusted logistic regression analysis of factors associated with breast exam uptake among women aged 15–49 years, CDHS 2021–2022 (N = 19,287).**

| Variable | | OR (95% CI) | P-value | AOR (95% CI) | P-value |
|---|---|---|---|---|---|
| Media Access | | | | | |
| | None | Ref. | | Ref. | |
| | One source | 1.38 (1.18–1.62) | <0.001 | **1.34 (1.13–1.58)** | **0.001** |
| | Two or more sources | 1.83 (1.40–2.40) | <0.001 | **1.47 (1.13–1.91)** | **0.004** |
| Smartphone Ownership | | | | | |
| | No phone | Ref. | | Ref. | |
| | Non-smartphone | 1.59 (1.19–2.12) | 0.002 | **1.35 (1.03–1.78)** | **0.032** |
| | Smartphone | 1.63 (1.33–1.99) | <0.001 | **1.37 (1.03–1.82)** | **0.028** |
| Internet use (last 12 months) | | | | | |
| | No | Ref. | | Ref. | |
| | Yes | 1.22 (1.04–1.44) | 0.016 | 1.10 (0.88–1.36) | 0.408 |
| Motorcycle Ownership | | | | | |
| | No | Ref. | | Ref. | |
| | Yes | 1.07 (0.84–1.36) | 0.581 | 0.79 (0.58–1.07) | 0.130 |
| Travel time to the facility | | | | | |
| | ≤10 minutes | Ref. | | Ref. | |
| | 11–30 minutes | 0.73 (0.63–0.85) | <0.001 | **0.84 (0.71–0.98)** | **0.026** |
| | >30 minutes | 0.37 (0.27–0.52) | <0.001 | **0.55 (0.39–0.78)** | **0.001** |
| Healthcare decision-making | | | | | |
| | Autonomous | Ref. | | Ref. | |
| | Joint decision | 0.76 (0.64–0.91) | 0.002 | **0.82 (0.70–0.96)** | **0.013** |
| | Not autonomous | 0.68 (0.51–0.90) | 0.007 | **0.70 (0.52–0.94)** | **0.018** |
| Women's mean (years) | | | | | |
| | 15-29 | Ref. | | Ref. | |
| | 30-39 | 2.96 (2.44–3.59) | <0.001 | **1.82 (1.49–2.23)** | **<0.001** |
| | 40-49 | 2.90 (2.37–3.53) | <0.001 | **2.04 (1.61–2.58)** | **<0.001** |
| Wealth index | | | | | |
| | Poorest | Ref. | | Ref. | |
| | Poorer | 1.04 (0.85–1.26) | 0.724 | 0.91 (0.73–1.13) | 0.387 |
| | Middle | 1.14 (0.92–1.41) | 0.225 | 0.94 (0.74–1.19) | 0.597 |
| | Richer | 1.44 (1.14–1.82) | 0.003 | 0.99 (0.76–1.30) | 0.967 |
| | Richest | 2.72 (2.12–3.47) | <0.001 | **1.86 (1.40–2.48)** | **<0.001** |
| Education level | | | | | |
| | No education | Ref. | | Ref. | |
| | Incomplete primary | 1.24 (0.98–1.57) | 0.067 | 1.11 (0.87–1.43) | 0.404 |
| | Complete primary | 1.30 (0.98–1.72) | 0.066 | 1.22 (0.88–1.70) | 0.227 |
| | Incomplete secondary | 1.08 (0.87–1.36) | 0.474 | 1.27 (0.97–1.67) | 0.087 |
| | Complete secondary | 1.50 (1.12–2.00) | 0.006 | 1.47 (1.04–2.08) | 0.030 |
| | Higher | 1.89 (1.35–2.64) | <0.001 | 1.51 (0.90–2.55) | 0.122 |
| Occupation | | | | | |
| | Not working | Ref. | | Ref. | |
| | Informal | 1.10 (0.91–1.32) | 0.335 | 1.06 (0.86–1.30) | 0.610 |
| | Professional/Formal | 2.04 (1.66–2.50) | <0.001 | 0.82 (0.67–1.01) | 0.067 |

*(Continued)*

**Table 2.** (Continued)

| Variable | | OR (95% CI) | P-value | AOR (95% CI) | P-value |
|---|---|---|---|---|---|
| Place of residence | | | | | |
| | Urban | Ref. | | Ref. | |
| | Rural | 0.64 (0.53–0.77) | <0.001 | 0.98 (0.82–1.18) | 0.861 |

**OR** = Unadjusted Odds Ratio; **AOR** = Adjusted Odds Ratio; **95% CI** = 95% Confidence Interval; **Ref.** = Reference group.

30–39, smartphone use was associated with increased screening (AOR = 1.71; 95% CI: 1.07–2.75; p = 0.026). However, no significant effect was observed among the youngest age group (15–29 years; AOR = 0.89; 95% CI: 0.54–1.46; p = 0.650) (S1 Table). In contrast, the interaction between media exposure and smartphone ownership (F(4, 657) = 0.63, p = 0.642), media exposure and education (F(10, 651) = 1.03, p = 0.417), and smartphone ownership and place of residence (F(2, 659) = 2.60, p = 0.075), they were not statistically significant, suggesting that their effects on breast cancer screening were independent (S1 Table).

## Discussion

This study highlights key determinants of breast exam uptake among women aged 15–49 in Cambodia, advancing current knowledge by demonstrating the significant associations of digital access, transportation, and women's autonomy with breast cancer screening using the most recent nationally representative data. Consistent with previous research from LMICs, media exposure was strongly associated with higher screening rates, with women exposed to two or more media sources having 47% higher odds of having had a breast cancer exam (AOR = 1.47, 95% CI: 1.13–1.91). In Cambodia, Mass media campaigns, including radio and television programs supported by the Ministry of Health and partners, likely play a key role in increasing awareness and encouraging early detection [13,17].

The finding that owning a non-smartphone was significantly associated with increased odds of undergoing a breast exam (AOR = 1.35; 95% CI: 1.03–1.78) suggests that even basic mobile phone access may facilitate communication about health services in Cambodia. Interestingly, smartphone ownership also showed a significant positive association (AOR = 1.37; 95% CI: 1.03–1.82). While smartphones provide broader access to health-related information and digital content, the similar effect sizes suggest that limited digital literacy and uneven adoption may constrain their potential, consistent with technology adoption frameworks in LMICs [24,25]. These findings align with regional evidence from Southeast Asia, where mobile phone access has been shown to enhance uptake of maternal and reproductive health services in resource-limited settings [26–30]. Expanding mobile health (mHealth) initiatives that are inclusive of both smartphone and non-smartphone users—particularly through voice and text-based interventions, as well as increasingly sophisticated yet user-friendly digital health content—could strengthen outreach and improve breast cancer screening coverage, especially among underserved populations in Cambodia [31]. Future research should explore not only device ownership but also women's digital literacy, usage patterns, and engagement with health-related content to better understand and leverage the role of mobile technology in preventive care.

Longer travel time to health facilities was strongly linked to lower breast exam uptake: women living more than 30 minutes away had 46% lower odds compared to those within 10 minutes (AOR = 0.54, 95% CI: 0.38–0.76). This supports the emphasis of Cambodia's National Strategic Plan for Cancer Control on decentralizing screening services to reduce travel burdens [12,13,17]. Despite progress, physical distance and transportation remain barriers, especially for poorer women. Women's autonomy in healthcare decision-making was positively associated with breast cancer screening; those making joint decisions had 19% lower odds (AOR = 0.81, 95% CI: 0.69–0.94), and those women not autonomous had 31% lower odds (AOR = 0.69, 95% CI: 0.51–0.94) compared to autonomous women. The result aligns with regional evidence that empowerment facilitates the use of preventive care [32–34]. Cambodia's community health worker programs aim to

enhance women's agency but may require scaling up. The wealthiest women had more than twice the odds of undergoing a breast exam compared to the poorest (AOR = 2.13, 95% CI: 1.60–2.84), reflecting persistent socioeconomic disparities despite health equity initiatives, such as the Health Equity Fund [7,35–40]. Education level and urban residence were not significantly associated after adjustment, indicating that access and empowerment may be stronger determinants in Cambodia's context. These findings reinforce the need to strengthen media campaigns, improve healthcare access, and empower women to reduce breast cancer screening inequalities in Cambodia [17,37].

## Policy implications

To address the persistently low rates of breast cancer screening among Cambodian women, particularly in rural and lower-income communities, a multifaceted approach is urgently needed. In the short term, digital health solutions—such as mobile health (mHealth) through messaging, appointment reminders, and awareness campaigns—can target digitally connected women. In contrast, Village Health Support Groups (VHSGs) and primary community health workers can provide education, referrals, and empower women to make autonomous health decisions [37,41]. In the medium term, targeted subsidies through mechanisms like the Health Equity Fund (HEF) can reduce financial barriers for the poorest households, and community-based outreach programs, including mobile clinics and local screening campaigns, can enhance accessibility in underserved areas [42–44]. Long-term strategies should focus on institutionalizing screening within the Cambodia National Cancer Control Plan (2025–2030), integrating digital health innovations into maternal and reproductive health programs, and establishing monitoring systems with measurable indicators [17]. By implementing prioritized, phased interventions with built-in evaluation, policymakers can effectively improve breast cancer screening uptake and equity across Cambodia.

## Limitations

This study has several limitations. The cross-sectional design limits the ability to infer causal relationships between determinants and ever having a breast exam. Data on ever having a breast exam, the possible influence of COVID-19 disruptions, and the self-reported data, which may introduce recall bias or social desirability bias. The outcome measure captured whether women had ever had a breast exam, which does not necessarily reflect current or regular screening behavior. Additionally, essential factors such as cultural beliefs, detailed knowledge about breast cancer, and quality of health services were not captured, potentially leading to residual confounding. Lastly, geographic accessibility was measured by women's self-reported travel time, which may be subject to perception bias, but it did not account for other access barriers, such as transportation costs or facility readiness.

## Conclusion

Breast exam uptake among Cambodian women remains low and inequitable. Digital access, transportation, and autonomy significantly influence screening behavior. This study makes a novel contribution by identifying emerging digital and empowerment factors associated with screening, using the most recent national data. Longitudinal and intervention studies are needed to evaluate scalable solutions for improving screening coverage and equity in Cambodia.

## Supporting information

**S1 Table. Adjusted Odds Ratios (AOR) for Breast Cancer Screening by Smartphone Ownership and Age Group (Interaction Model).**
(DOCX)

## Acknowledgments

We thank the Ministry of Planning and the Ministry of Health of Cambodia for granting access to the CDHS dataset. We also acknowledge ICF International for technical support in survey implementation.

## Author contributions

**Conceptualization:** Samnang Um, Channnarong Phan, Daraden Vang, Tharuom Ny, Sothy Heng.

**Data curation:** Samnang Um, Tharuom Ny.

**Formal analysis:** Samnang Um, Channnarong Phan, Daraden Vang.

**Investigation:** Samnang Um.

**Methodology:** Samnang Um, Channnarong Phan, Daraden Vang, Tharuom Ny, Sothy Heng.

**Project administration:** Samnang Um.

**Resources:** Samnang Um.

**Software:** Samnang Um.

**Supervision:** Samnang Um, Tharuom Ny, Sothy Heng.

**Validation:** Samnang Um.

**Visualization:** Samnang Um.

**Writing – original draft:** Samnang Um, Channnarong Phan, Daraden Vang, Tharuom Ny, Sothy Heng.

**Writing – review & editing:** Samnang Um, Tharuom Ny, Sothy Heng.

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
