## [Decision Letter · Decision Letter 0]

16 Aug 2025

PDIG-D-25-00512Digital Access, Transportation, and Women’s Empowerment in Breast Cancer Screening Uptake Among Cambodian Women: Analysis of the Cambodia Demographic and Health Survey 2021-2022PLOS Digital Health Dear Dr. Um, Thank you for submitting your manuscript to PLOS Digital Health. After careful consideration, we feel that it has merit but does not fully meet PLOS Digital Health's publication criteria as it currently stands. Therefore, we invite you to submit a revised version of the manuscript that addresses the points raised during the review process. Please submit your revised manuscript within 60 days Oct 15 2025 11:59PM. If you will need more time than this to complete your revisions, please reply to this message or contact the journal office at digitalhealth@plos.org. Please include the following items when submitting your revised manuscript:* A rebuttal letter that responds to each point raised by the editor and reviewer(s). You should upload this letter as a separate file labeled 'Response to Reviewers '. This file does not need to include responses to any formatting updates and technical items listed in the 'Journal Requirements' section below.* A marked-up copy of your manuscript that highlights changes made to the original version. You should upload this as a separate file labeled 'Revised Manuscript with Track Changes '.* An unmarked version of your revised paper without tracked changes. You should upload this as a separate file labeled 'Manuscript '. If you would like to make changes to your financial disclosure, competing interests statement, or data availability statement, please make these updates within the submission form at the time of resubmission. Guidelines for resubmitting your figure files are available below the reviewer comments at the end of this letter. We look forward to receiving your revised manuscript. Kind regards, Haleh AyatollahiSection EditorPLOS Digital Health Haleh AyatollahiSection EditorPLOS Digital Health Leo Anthony CeliEditor-in-ChiefPLOS Digital Healthorcid.org/0000-0001-6712-6626 **Journal Requirements:**1. Please provide an Author Summary. This should appear in your manuscript between the Abstract (if applicable) and the Introduction, and should be 150–200 words long. The aim should be to make your findings accessible to a wide audience that includes both scientists and non-scientists. Sample summaries can be found on our website under Submission Guidelines: 

https://journals.plos.org/digitalhealth/s/submission-guidelines#loc-parts-of-a-submission 2. If the reviewer comments include a recommendation to cite specific previously published works, please review and evaluate these publications to determine whether they are relevant and should be cited. There is no requirement to cite these works unless the editor has indicated otherwise.  **Additional Editor Comments (if provided):****Reviewers' Comments:** Reviewer's Responses to Questions

**Comments to the Author**

1. Does this manuscript meet PLOS Digital Health’s publication criteria ? Is the manuscript technically sound, and do the data support the conclusions? The manuscript must describe methodologically and ethically rigorous research with conclusions that are appropriately drawn based on the data presented.

Reviewer #1: Yes

Reviewer #2: Yes

Reviewer #3: Yes

2. Has the statistical analysis been performed appropriately and rigorously?

Reviewer #1: Yes

Reviewer #2: Yes

Reviewer #3: Yes

3. Have the authors made all data underlying the findings in their manuscript fully available (please refer to the Data Availability Statement at the start of the manuscript PDF file)?

Reviewer #1: Yes

Reviewer #2: Yes

Reviewer #3: Yes

4. Is the manuscript presented in an intelligible fashion and written in standard English?

Reviewer #1: Yes

Reviewer #2: Yes

Reviewer #3: Yes

5. Review Comments to the Author

Reviewer #1: Comments to the authors:

I suggest clarifying why the study utilized data from the Cambodia Demographic and Health Survey (CDHS) 2021–2022 years exactly. Is it the only available dataset or the last available dataset? Or do you choose it for another reason?

In the introduction, you mentioned The National Action Plan for Cervical Cancer 56 Prevention and Control (2019-2023), lines 55-56; I suggest also adding the results or recommendations from this plan.

Reviewer #2: This study examines the determinants of breast exam uptake among Cambodian women aged 15–49 using nationally representative data from the Cambodia Demographic and Health Survey (2021–2022). The authors focus on factors such as media exposure, digital access (including mobile phone and internet use), travel time to health facilities, women’s autonomy in healthcare decisions, socioeconomic status, education, and place of residence. Using survey-weighted logistic regression, they identify significant associations between these factors and breast exam uptake, highlighting persistent disparities related to access, empowerment, and wealth.

The manuscript is timely and adds value to exsiting literature. The discussion section is thoughtfully developed and effectively contextualizes the findings within Cambodia’s health system and broader regional evidence. The authors demonstrate a strong grasp of underlying barriers such as digital literacy and transportation challenges, while linking results to national health initiatives. The inclusion of a policy implications subsection is a valuable addition.

A few suggestions are listed below to further tighten the study.

Major

=======

1.The introduction sets a solid foundation by outlining the public health importance and Cambodia’s context. However, it would benefit from improved flow and focus. Currently, it is somewhat repetitive and data-heavy, particularly the transition from global to national statistics, where percentages alone would suffice. Some new topics (like media exposure and motorcycle ownership) appear abruptly without clear links to prior content. Paragraphs tend to mix multiple ideas, making the core message harder to follow. To strengthen the narrative, I recommend a clearer funnel structure: starting broad, then narrowing to national context, screening barriers, potential solutions, research gaps, and finally study aims, ensuring smooth transitions between each section.

2.To improve flow in Introduction, consider adding transitional or framing sentences at the start of thematic paragraphs in the Introduction. For example: “In addition to informational and economic barriers, physical access remains an important but understudied determinant of screening uptake in Cambodia…” This would help readers understand the rationale for introducing each factor.

3.Independent Variables:

3(a) The description of independent variables would benefit from clearer organization to enhance readability and comprehension. I recommend structuring them as distinct subsections or using clear formatting cues (e.g., bolded subheadings) with complete sentences to delineate each variable or concept.

3(b) Explicitly stating that independent variables are derived from standardized CDHS survey questions to avoid confusion. For example:

Transportation: It was derived from CDHS survey question whether the household owned a motorcycle.

3(c) The description of "Digital Access" in the methods section currently groups mobile phone ownership, internet use, and media exposure together, which reads like a composite or latent variable. Since these are separate independent variables in the analysis, describing each variable individually and avoiding wording that implies they form a combined index would be helpful.

4.If using survey-weighted logistic regression (as indicated by the design-based F-statistics), please add a brief note explaining why F-statistics are reported instead of the usual χ² tests (OR) include “adjusted for survey design” to clarify this for readers.

5.Only two interaction terms (age × smartphone ownership and media exposure × smartphone ownership) were tested, without a stated rationale for their selection. Given the conceptual focus on information access, socioeconomic status, and structural barriers, other plausible interactions should be considered. for example, wealth × media exposure, education × media exposure, or rural residence × phone ownership. If these were examined but found non-significant, this should be stated for transparency. If not examined, authors should provide a rationale for limiting the analysis to the current two interactions.

Minor

=====

1.Discussion is overall very good. Minor tightening of language to avoid redundancies and clarity on the p-value explanation would make it solid.

Eg: lines 310-313 revision suggestion can be “Although the adjusted odds ratios (AORs) for non-smartphone and smartphone ownership are similar—indicating comparable effect sizes, the difference in statistical significance (p-values) might reflect differences in sample size or variability. Beyond statistical factors, this similarity in effect size despite differing p-values may also point to real-world challenges in Cambodia, such as limited digital literacy, unequal access to health apps, and a lack of tailored digital health campaigns that fully harness smartphone capabilities for breast cancer screening.”

2.Table 2 has “Ref.” in multiple places, but non-specialist readers might not notice what each variable is compared against. A footnote could help.

3.Consider adding bivariant correlation table - a compact table makes it verifiable and easier to compare with adjusted results.

Reviewer #3: The manuscript has been meticulously crafted to meet the publication standards of PLOS Digital Health. It presents technically robust findings that substantiate the conclusions drawn. The methodology is clearly outlined, demonstrating the ethical rigor of the research, and the conclusions are appropriately derived from the data presented. Furthermore, the statistical analysis is thorough, with results clearly presented in tables. The manuscript is well-written in clear, concise English, ensuring ease of understanding for readers.

COMMENTS TO THE AUTHORS

Abstract

Strengths

The abstract succinctly summarizes the background, methods, results, and conclusions. The use of statistical results (AOR, 95% CI) improves scientific rigor.

Suggestions for improvement

In the abstract, consider clarifying the study design used in the first sentence of the methods to set expectations. Avoid redundancy in the first sentence ("The breast cancer cases are increasing…" could be rephrased for smoother readability, e.g., "Breast cancer incidence is increasing…"). Specify in the abstract whether “breast examination” mainly refers to clinical breast examination or includes mammography; this will reduce ambiguity for readers unfamiliar with the CDHS measurement approach. Streamline the interpretation in the last part of the abstract by focusing on actionable implications rather than repeating results.

Introduction

Strengths

The introduction provides strong epidemiological context for both global and Cambodian breast cancer burden. Logical flow from general problem to national context, and then to specific gaps in knowledge. Inclusion of references to national strategies strengthens the policy relevance.

Suggestions for improvement

The introduction is quite dense; consider shortening sentences to improve readability for an international audience. Clarify how your study builds upon and differs from the previous CDHS analysis cited in [14], be explicit about the novelty. When discussing digital access and transportation, briefly highlight any hypothesized pathways (e.g., improved access to information, reduced travel barriers) to strengthen the conceptual framework. Consider integrating a short paragraph that synthesizes these factors into a clear conceptual model leading to the study objectives.

Methods

Strengths

Ethical approval and consent procedures are clearly stated. The sampling design is described in detail, including weights, clustering, and stratification. The outcome and independent variables are well-defined with operational categories.

Suggestions for improvement

The description of variables and survey design is thorough. Yet, details on how missing data were handled (besides exclusions) could be expanded.

Add clarity on whether all included variables were pre-specified or selected based on bivariate significance.

Explain if ethical approval was needed for secondary data use, even if waived.

Clarify whether “breast examination” responses included both provider-led CBEs and mammography, and whether the survey instrument differentiated them. Specify how missing data were handled beyond the initial exclusion (e.g., for covariates — was complete case analysis used?). Justify why certain potential confounders (e.g., marital status, parity) were excluded, given their known relevance in screening literature. When describing effect modification testing, clarify why age × smartphone ownership was chosen and not tested for other plausible interactions (e.g., media exposure × residence). The Hosmer-Lemeshow test with survey data has limitations — you could note this to preempt reviewer concerns.

Results

Strengths

Results are clearly structured, moving from descriptive statistics to bivariate associations and multivariable analysis. Tables are detailed and provide both unadjusted and adjusted estimates. Reporting of AORs with CIs and p-values is consistent.

Suggestions for improvement

In the descriptive section, consider adding a brief narrative interpretation (e.g., “Most women owned a smartphone, yet only ~11% had been screened…”). Highlight notable unexpected findings in-text (e.g., higher screening uptake in rural vs. urban women in unadjusted analysis) and signal that these are further explored in discussion.

In the multivariable analysis, clarify the rationale for including interaction terms and report model diagnostics (e.g., AUC) more prominently.

Explain why the ROC AUC threshold of 0.653 is described as “moderate” and its implications for the model’s predictive capacity.

Discussion

Strengths

Interprets findings in the context of Cambodian health policy and global evidence. Thoughtful consideration of both digital and traditional media access as enablers of screening. Addresses the socioeconomic gradient in screening uptake.

Suggestions for improvement:

The discussion is detailed but could benefit from a more concise synthesis of the main findings before moving into interpretations. It should better emphasize how the findings advance current knowledge. Some interpretations (e.g., smartphone vs. non-smartphone similarity) could be expanded by referencing relevant behavioral or technological adoption theories (referencing digital literacy issues more deeply). The rural–urban pattern reversal in unadjusted results warrants brief exploration, even if it becomes non-significant after adjustment — possible role of community outreach or health equity programs? When suggesting mHealth interventions, be explicit about feasibility in the Cambodian context given varying digital literacy levels. Link findings more directly to specific elements of the National Cancer Control Plan and other ongoing initiatives.

Policy Implications

Strengths

Well-linked to existing national programs like the Health Equity Fund and VHSG networks. Integrates digital strategies with community-based approaches.

Suggestions for improvement:

Consider prioritizing recommendations (short-, medium-, and long-term) for clearer applicability. Specify how proposed strategies might be evaluated for impact.

Limitations

Strengths

Addresses key methodological constraints (cross-sectional design, recall bias, lack of differentiation in screening type). Acknowledges unmeasured confounders.

Suggestions for improvement

Explicitly note that the outcome measure captures “ever” having a breast exam, which may not reflect current or regular screening behavior. Mention the possible influence of COVID-19 service disruptions during the data collection period (Sept 2021–Feb 2022) as a contextual limitation. Acknowledge that travel time as a self-reported measure may be subject to perception bias.

Conclusion

Strengths

Summarizes the main determinants concisely. Reinforces the policy relevance of the findings.

Suggestions for improvement

Add one sentence clearly stating the novel contribution of the study to the literature. Consider ending with a forward-looking perspective (e.g., need for longitudinal or intervention studies to address gaps).

6. PLOS authors have the option to publish the peer review history of their article (what does this mean? ). If published, this will include your full peer review and any attached files.

**Do you want your identity to be public for this peer review?** For information about this choice, including consent withdrawal, please see our Privacy Policy .

Reviewer #1: **Yes: ** Jehad Omar Abualrob

Reviewer #2: **Yes: ** Parvati Naliyatthaliyazchayil

Reviewer #3: **Yes: ** Kinene Andrew

---

## [Decision Letter · Decision Letter 1]

3 Sep 2025

Digital Access, Transportation, and Women’s Empowerment in Breast Cancer Screening Uptake Among Cambodian Women: Analysis of the Cambodia Demographic and Health Survey 2021-2022

PDIG-D-25-00512R1

Dear Dr. Um,

We're pleased to inform you that your manuscript has been judged scientifically suitable for publication and will be formally accepted for publication once it meets all outstanding technical requirements.

Within one week, you'll receive an e-mail detailing the required amendments. When these have been addressed, you'll receive a formal acceptance letter and your manuscript will be scheduled for publication.

An invoice for payment will follow shortly after the formal acceptance. To ensure an efficient process, please log into Editorial Manager at https://www.editorialmanager.com/pdig/ click the 'Update My Information' link at the top of the page, and double check that your user information is up-to-date. For billing related questions, please contact billing support at https://plos.my.site.com/s/.

Kind regards,

Haleh Ayatollahi

Section Editor

PLOS Digital Health

Additional Editor Comments (optional):

Reviewer #1: 

Reviewer #2: 

Reviewer #3:

Reviewers' comments:

Reviewer's Responses to Questions

**Comments to the Author**

1. If the authors have adequately addressed your comments raised in a previous round of review and you feel that this manuscript is now acceptable for publication, you may indicate that here to bypass the “Comments to the Author” section, enter your conflict of interest statement in the “Confidential to Editor” section, and submit your "Accept" recommendation.

Reviewer #1: All comments have been addressed

Reviewer #2: All comments have been addressed

Reviewer #3: All comments have been addressed

2. Does this manuscript meet PLOS Digital Health’s publication criteria ? Is the manuscript technically sound, and do the data support the conclusions? The manuscript must describe methodologically and ethically rigorous research with conclusions that are appropriately drawn based on the data presented.

Reviewer #1: Yes

Reviewer #2: Yes

Reviewer #3: Yes

3. Has the statistical analysis been performed appropriately and rigorously?

Reviewer #1: Yes

Reviewer #2: Yes

Reviewer #3: Yes

4. Have the authors made all data underlying the findings in their manuscript fully available (please refer to the Data Availability Statement at the start of the manuscript PDF file)?

Reviewer #1: Yes

Reviewer #2: Yes

Reviewer #3: Yes

5. Is the manuscript presented in an intelligible fashion and written in standard English?

PLOS Digital Health does not copyedit accepted manuscripts, so the language in submitted articles must be clear, correct, and unambiguous. Any typographical or grammatical errors should be corrected at revision, so please note any specific errors here.

Reviewer #1: Yes

Reviewer #2: Yes

Reviewer #3: Yes

6. Review Comments to the Author

Please use the space provided to explain your answers to the questions above. You may also include additional comments for the author, including concerns about dual publication, research ethics, or publication ethics. (Please upload your review as an attachment if it exceeds 20,000 characters)

Reviewer #1: Great job

Best wishes

Reviewer #2: All my comments have been addressed by authors.

Reviewer #3: The authors have satisfactorily addressed all the comments. The manuscript is technically sound, well-written in standard English, and supported by appropriate data that substantiate the conclusions. I am confident that it meets PLOS Digital Health’s publication criteria and is ready for publication.

7. PLOS authors have the option to publish the peer review history of their article (what does this mean? ). If published, this will include your full peer review and any attached files.

**Do you want your identity to be public for this peer review?** For information about this choice, including consent withdrawal, please see our Privacy Policy . 

Reviewer #1: Yes: Jehad Omar Abualrob

Reviewer #2: Yes: Parvati Naliyatthaliyazchayil

Reviewer #3: Yes: Kinene Andrew
